# The Influence of Various Forms of Nitrogen Fertilization and Meteorological Factors on Nitrogen Compounds in Soil under Laboratory Conditions

**Rūta Dromantienė [1], Irena Pranckietienė [1], Darija Jodaugienė [1],\* and Aurelija Paulauskienė [2]**

[1] Institute of Agroecosystems and Soil Sciences, Vytautas Magnus University Agriculture Academy Studentų 11, Akademija, LT-53361 Kaunas, Lithuania; ruta.dromantiene@vdu.lt (R.D.); irena.pranckietiene@vdu.lt (I.P.)

[2] Institute of Agricultural and Food Sciences, Vytautas Magnus University Agriculture Academy Studentų 11, Akademija, LT-53361 Kaunas, Lithuania; aurelija.paulauskiene@vdu.lt

\* Correspondence: darija.jodaugiene@vdu.lt

**Abstract:** Nitrogen is one of the main factors that shapes soil fertility and the productivity of crops, although its abundance can also cause damage to the environment. The aim of this study is to evaluate the influences of different forms of nitrogen fertilizers, soil temperature, and precipitation on the changes of nitrogen compounds ($N-NH_4^+$, $N-NO_3^-$, and $N_{min}$) in two soil layers. Two pot experiments are performed, involving simulated precipitation levels of 10- and 20 mm. Urea and ammonium nitrate fertilizers are used for fertilization. The soil samples are stored in pots in a climate chamber at different temperatures of 5, 10, 15, and 20 °C. After seven days, the changes of nitrogen compounds ($N-NH_4^+$, $N-NO_3^-$, and $N_{min}$) in 0–15 and 15–30 cm soil layers are analyzed. This study shows that the amount of $N-NH_4^+$ nitrogen in the soil depends on the fertilizer form and soil temperature. In the temperature range of 5–20 °C, significantly more $N-NH_4^+$ nitrogen is present in urea-fertilized soil. The migration of $N-NH_4^+$ into the deeper 15–30 cm soil layer at both the 10- and 20-mm simulated precipitation levels is negligible. The $N-NO_3^-$ contents in the 0–15 cm soil layer in the temperature range of 5–20 °C are 1.7–2.3 times lower in the urea-fertilized soil than in the ammonium nitrate-fertilized soil at a 10-mm simulated precipitation level and 1.6–2.2 times lower at 20 mm. The $N_{min}$ contents in soil are directly dependent on the fertilizer form and soil temperature for both levels of simulated precipitation.

**Keywords:** nitrogen; ammonium nitrate; urea; temperature; soil moisture

## 1. Introduction

Anthropogenic activities have doubled the amount of reactive nitrogen circulating on Earth, leading to excess nutrient transfer to freshwater ecosystems and ultimately causing the eutrophication of freshwater and marine aquatic ecosystems [1,2]. Agriculture is considered one of the sources of nitrogen pollution (ammonia ($NH_3$) and nitrous oxide ($N_2O$) to air, and nitrate ($N-NO_3^-$) to surface and ground water) [3–5]. The adoption of more stringent environmental standards and increasing nitrogen fertilizer prices oblige researchers, fertilizer manufacturers, and agricultural producers to look for rational and sustainable solutions to reduce nitrogen emissions [5–7], which is also directly related to nitrogen losses from mineral fertilizers. Mineral nitrogen losses from the soil are caused by the unbalanced and untimely fertilization of crops, improper choices for fertilizer forms, and other factors [8–12]. Nitrogen losses from the soil and mineral nitrogen fertilizers directly depend on the intensity of nitrogen compound transformation in the soil. It has been documented that the changes of

the ammonium nitrogen and nitrate nitrogen forms depend on the crop growth season, fertilization intensity, fertilizer form, precipitation rate, soil temperature, etc. [11–13]. Precipitation directly alters N cycling and transformation [14]. Changes in these rates are also influenced by the soil temperature, because it affects the activity of urease, nitrifier communities, and nitrification rate in the soil [15,16]. For example, some studies have identified the stimulating effects of soil temperature [17,18], whereas a few other studies have shown that warming can suppress ammonification [19] or that the temperature dependency of ammonification is not significant [20]. According to Russell et al. [21] for sandy soil, the maximum rates of nitrification decreased as the temperature was reduced from 20 °C to 5 °C. Tripolskaja and Verbylienė [22] suggest that precipitation during the warm period of the year promotes the leaching and migration of chemical elements from topsoil.

Nitrates are especially dangerous to the environment because they are not absorbed by the soil sorption complex, meaning that excessive amounts of nitrates may be leached [23,24]. Haberle et al. [25] found that about 35–70% of nitrates can be leached into soil layers deeper than 1 m in certain conditions. Gallucci et al. [26] showed that the losses of N from fertilizer range from 20 to 30%, depending of the N sources, soil moisture, and pH. In recent years, N losses from fertilization practices have become an issue due to increased nitrate leaching [27].

Our study hypothesizes that research into the changes in ammonium nitrogen ($N-NH_4^+$), nitrate nitrogen ($N-NO_3^-$), and mineral nitrogen forms ($N-NH_4^+$ and $N-NO_3^-$, hereinafter $N_{min}$) in soil, resulting from the complex effects of fertilizer forms, soil temperatures, and precipitation, will provide knowledge and contribute to a deeper understanding about mineral nitrogen dynamics in soil and crop fertilization modeling possibilities to ensure optimal nitrogen supply for crops and minimize environmental nitrogen pollution in warming climatic conditions.

Specifically, the current study aims to evaluate the impact of different forms of nitrogen fertilizers, soil temperature, and precipitation levels on the changes of nitrogen compounds ($N-NH_4^+$, $N-NO_3^-$, and $N_{min}$) at soil layers of 0–15 and 15–30 cm.

## 2. Materials and Methods

### 2.1. Study Site and Experimental Design

Two two-factor pot experiments that differed in the amount of simulated precipitation were conducted at Vytautas Magnus University Agriculture Academy (Lithuania) in 2018. Over a period of 7 days, the first pot received a simulated amount of precipitation of 10 mm, and the second pot received 20 mm. The experiments were replicated twice in the same year and conditions. In terms of the two-factor design, factor A included different forms of fertilizers, namely, a control (without fertilizers), ammonium nitrate ($NH_4NO_3$), and urea ($CO(NH_2)_2$. For factor B, this included varying soil temperatures of 5, 10, 15, and 20 °C. The experimental treatments were arranged in four replicates with a completely randomized block design.

The experiment involved the simulation of soil temperature and moisture, which are relevant factors during plant growth and the fertilization period. The experiments were carried out in a controlled climate chamber (RUMED 1301). The soil from the 0–30 cm layer of *Luvisol* soil [28] was used, characterized by a loamy texture (15.1% clay, 44.5% silt, and 40.4% sand). Before the experiment, the $pH_{KCl}$ value of the topsoil was determined to be 6.8, the concentration of available $P_2O_5$ was 162 mg kg$^{-1}$, the available $K_2O$ was 188 mg kg$^{-1}$, and the organic carbon content was 1.8%. The mineral nitrogen content was 9.8 mg kg$^{-1}$ ($N-NH_4^+$—0.72 mg kg$^{-1}$; $N-NO_3^-$—9.08 mg kg$^{-1}$). The soil was homogenized and moistened to a 25% humidity level, with a water capacity of 46.2%.

The experimental soil samples were placed into cylindrical pots that were 10.5 cm in diameter and 30 cm in height. The formed soil density was close to the equilibrium density of the loam of 1.40 Mg m$^{-3}$. Nitrogen fertilizers of different forms were applied to the soil surfaces of each pot. The nitrogen rate was 90 kg ha$^{-1}$. The experiment was performed for 7 days at 70% air humidity. Simulated precipitation levels of 10 and 20 mm were applied in equal parts on the first and fifth days after fertilizer application.

The soil moisture levels were the following: (1) Before the first irrigation—25.0%; after the first irrigation—26.7% (precipitation of 10 mm) and 28.3% (precipitation of 20 mm); (2) before the second irrigation—24.0–26.0% (precipitation of 10 mm) and 25.6–27.6% (precipitation of 20 mm); after the second irrigation—25.7–27.7% (precipitation of 10 mm) and 28.9–30.9% (precipitation of 20 mm).

The distribution of simulated precipitation (water) was performed manually using a *Stihl Sg 11* sprayer, having estimated the volume of water sprayed with one click. Watering was provided early in the morning. All pots received the same volume of water per event. The volume of water applied was determined according to the average meteorological data of the second to third ten-day period of March and the first ten-day period of April in the 2013–2017 period (Kaunas meteorological station, Lithuania 2013–2017), i.e., the beginning of the winter wheat and winter oilseed rape growing season in the northern part of the temperate climate zone (coordinates 54°53′3.26″ north latitude, 23°50′33.25″ east longitude), which is when the highest rates of nitrogen are used.

On the eighth day after the start of the experiment, the soil samples were taken from the 0–15 and 15–30 cm soil layers for analysis and the determination of the $N-NH_4^+$, $N-NO_3^-$, and $N_{min}$ contents.

## 2.2. Analytical Methods

Soil pH, available $P_2O_5$ and $K_2O$, the concentration of organic C and N, and contents of mineral nitrogen ($N-NH_4^+$ and $N-NO_3^-$) were analyzed in the Agrochemical Research Laboratory of the Lithuanian Research Centre for Agriculture and Forestry.

The topsoil was analyzed for $pH_{KCl}$ as measured in 1N KCl extractions (potentiometric method). The available phosphorus ($P_2O_5$) and available potassium ($K_2O$) were measured by the Egner–Riehm–Domingo (A-L) method, and organic carbon (C) was determined by the wet oxidation method. The percentages of sand, silt, and clay (soil texture) were determined using the pipette method. The $N_{min}$ contents in the soil samples were determined with two methods, namely, $N-NH_4^+$ by the spectrophotometric method, and $N-NO_3^-$ by the ionometric method. The chemicals used in this study were of an analytical grade. Chemical analyses were performed in three replications.

## 2.3. Statistical Analysis

A two-way statistical analysis of the experimental data was carried out with *TIBCO Statistica* version 7 (TIBCO Software, USA). The research data were evaluated using an analysis of variance (ANOVA) method. The significance of data was determined by Fisher's least significant difference (LSD) test with a significance level of $p \leq 0.05$. There were significant interactions between different forms of fertilizers (factor A) and soil temperatures (factor B), and the averages of factors A and B are not presented. Correlation and regression analyses were performed to determine the strengths and characters of the relationships between variables at a probability level of 95% [29,30].

## 3. Results and Discussion

The concentrations of $N_{min}$ in soil solutions are variable and dynamic [31]. Li et al. [32] suggested that mineral nitrogen rates are positively correlated with soil temperature and moisture levels. Guntinas et al. [33] reported that 25 °C was the maximum value for sensitivity to temperature. The data from this study define the variation of $N-NH_4^+$ nitrogen and $N-NO_3^-$ contents in the arable layer of soil of a certain temperature when fertilizing with different forms of nitrogen fertilizers at the precipitation levels of 10 and 20 mm.

Experiment 1. In the 0–15 cm soil layer, under the simulated precipitation level of 10 mm, the experimental findings show significantly higher amounts of $N-NH_4^+$ nitrogen in the urea-fertilized soil compared with the ammonium nitrate-fertilized soil (Figure 1). This was due to the amide nitrogen (hereinafter $N-NH_2^-$) form present in urea, part of which was transformed into $N-NH_4^+$, and a small part was converted into $N-NO_3^-$ within 7 days of the experiment. This is based on the fact that the rapid hydrolysis of urea in the soil can result in a high content of $N-NH_4^+$ ions, which slows down the nitrification process [34]. The scientific literature indicates that the transformation of $N-NH_4^+$ into

N-NO$_3$$^-$ can take about one week (at a temperature of 23 °C) [35]. Studies by Canadian researchers [36] show that nitrification is limited at 4 °C, extensive at 9 °C, and essentially complete after 48–68 days at 18 °C.

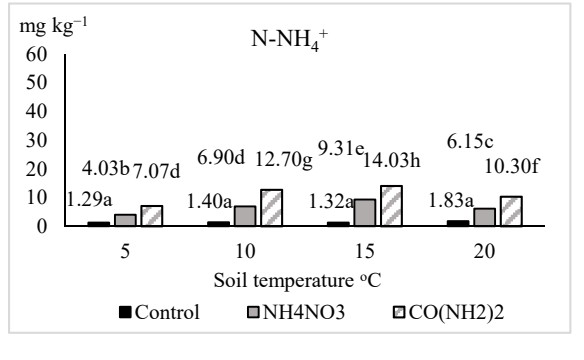
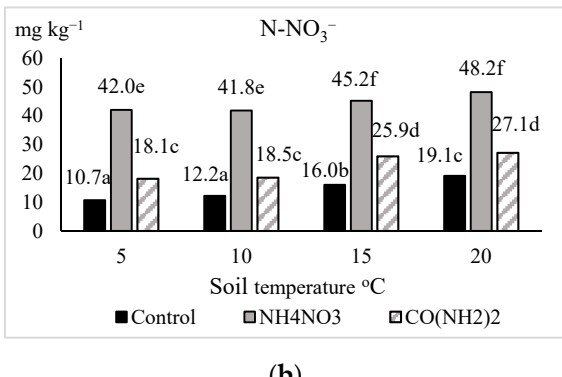

(**a**)                                                                 (**b**)

**Figure 1.** The influences of nitrogen fertilizer forms and ambient temperatures on the changes of (**a**) N-NH$_4$$^+$ and (**b**) N-NO$_3$$^-$ in the 0–15 cm soil layer with a simulated precipitation of 10 mm. Values followed by different letters are significantly different ($p \leq 0.05$) based on Fisher's LSD test.

As the soil temperature increased from 5 °C to 15 °C, both the N-NH$_4$$^+$ and N-NO$_3$$^-$ levels in the soil consistently increased, regardless of the form of the fertilizer.

For urea fertilization, the highest N-NH$_4$$^+$ content and significant increase was determined at a 15 °C soil temperature. This was likely determined by an increase in both the mineralization of organic matter and the intensity of hydrolysis of N-NH$_2$$^-$ at a higher soil temperature. This agrees with the research data documented by Rochette et al. [37], Abalos et al. [38], and Zhang et al. [12], which suggest that both increased intensive organic matter mineralization and the activity of *Uro* bacteria are associated with a higher soil temperature.

In our experiment, at a 20 °C soil temperature, the content of N-NH$_4$$^+$ was significantly lower than that at 15 °C. This difference may have been determined by increased nitrogen losses due to volatilization in the form of ammonia (NH$_3$) and a more intensive nitrification process. The second statement is corroborated by the data of this study, where the content of N-NO$_3$$^-$ at 20 °C was 4.6% higher than at 15 °C.

When fertilizing with ammonium nitrate, the highest N-NH$_4$$^+$ content in the soil was also found at 15 °C. In this case, the increase in the N-NH$_4$$^+$ content can be attributed to the more intensive release of mineral nitrogen compounds due to the naturally occurring processes related to soil organic matter mineralization [39,40] and the less intensive nitrification process than at 20 °C. This is supported by the data of our study, as a significant decrease in N-NH$_4$$^+$ and an increase of N-NO$_3$$^-$ were recorded at 20 °C soil temperature (Figure 1). The reduction in N-NH$_4$$^+$ content at 20 °C can also be attributed to the higher possibility for nitrogen loss due to volatilization, as research shows that nitrogen loss from ammonium nitrate when applied to the soil surface (broadcast) can be up to 2% [41]. The lowest amounts of N-NH$_4$$^+$ were present in the soil at 5 °C. Changes in the N-NH$_4$$^+$ content when fertilizing with ammonium nitrate were similar to those obtained with urea fertilization.

The regression analysis of the experimental data shows that in the 0–15 cm layer, the dependence of N-NH$_4$$^+$ on soil temperature (in the range of 5–20 °C) for both urea- and ammonium nitrate-fertilized soil can be described by the following quadratic equations: $y_{CO(NH2)2} = -3.424 + 1.683x - 0.093x^2$ and $y_{NH4NO3} = -3.135 + 1.6837x - 0.0603x^2$. In both cases, the relationships between the indicators were very strong ($R^2 = 0.980$ and $R^2 = 0.902$, respectively) and significant ($p \leq 0.05$).

The intensity of transformation of low-mobility N-NH$_4$$^+$ due to nitrification into very mobile N-NO$_3$$^-$ has an influence on nitrogen fertilizer use efficiency and environmental quality, especially in cases when the likelihood of N-NO$_3$$^-$ losses due to leaching is high and plant uptake is low (i.e., due to small root systems, high fertilizer rates, or excess rainfall). In urea-fertilized soil, the amounts of

N-NO$_3^-$ in the 0–15 cm layer, regardless of the soil temperature, were significantly lower and were 1.7–2.3 times lower compared to the ammonium nitrate-fertilized soil (Figure 1). This was largely determined by the composition of the fertilizer and the longer time taken for the conversion of N-NH$_2^-$ to N-NO$_3^-$. The experimental data show that the soil temperatures of 15 °C and 20 °C had the greatest influence on the amount of nitrate nitrogen, both when fertilizing with ammonium nitrate and urea. In these cases, the N-NO$_3^-$ levels in the soil were significantly higher than at 5 °C and 10 °C.

The change of N-NO$_3^-$ as influenced by soil temperature (in the range of 5–20 °C) can be described by the following linear equations: $y_{CO(NH2)2} = 14.377 + 0.617x$ ($R^2 = 0.828$; $p \leq 0.05$) and $y_{NH4NO3} = -38.8 + 0.44x$ ($R^2 = 0.960$; $p \leq 0.05$).

The N-NH$_4^+$ contents in the 15–30 cm soil layer in all cases were significantly lower than those in the 0–15 cm soil layer (Figures 1 and 2). The correlation analysis of the experimental data suggests that at the simulated precipitation level of 10 mm, there were no significant ($p \leq 0.05$) relationships between N-NH$_4^+$ content in the 0–15 and 15–30 cm layers. The low migration may have been caused by the ability of soil to absorb N-NH$_4^+$ ions [42], peculiarities of nitrogen compounds transformation at different temperatures [43], low infiltration of precipitation, and other factors [22].

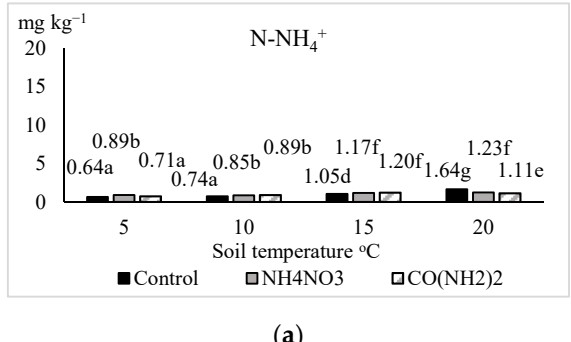

(a)

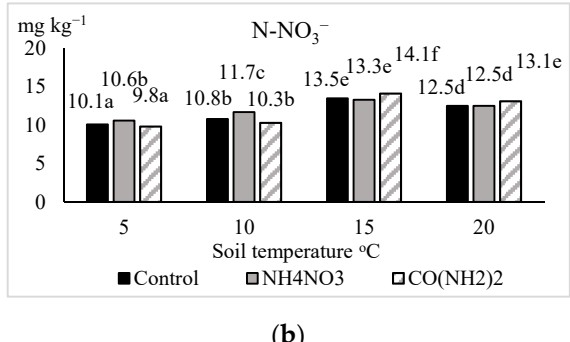

(b)

**Figure 2.** The influence of nitrogen fertilizers and ambient temperature on the changes of (**a**) N-NH$_4^+$ and (**b**) N-NO$_3^-$ in the 15–30 cm soil layer with a simulated precipitation amount of 10 mm. Values followed by different letters are significantly different ($p \leq 0.05$) based on Fisher's LSD test.

When fertilizing with urea, N-NH$_4^+$ levels in the 15–30 cm soil layer increased significantly with an increasing temperature (Figure 2); however, the maximum was reached at a soil temperature of 15 °C. Similar data were obtained in the 0–15 cm soil layer (Figure 1).

In the 15–30 cm soil layer, ammonium nitrate fertilization tended to increase the N-NH$_4^+$ content with an increasing soil temperature up to 20 °C, while urea fertilization tended to increase the N-NH$_4^+$ content with an increasing soil temperature up to 15 °C. Significantly higher levels of this compound were found in the soil at 15 and 20 °C compared with those present at 5 and 10 °C.

In this soil layer, both with urea and ammonium nitrate fertilization, the correlations between N-NH$_4^+$ and soil temperature were very strong ($R^2 = 0.904$ and $R^2 = 0.827$, respectively), but not significant ($p \geq 0.05$).

The content of N-NO$_3^-$ in the 15–30 cm soil layer increased with an increasing soil temperature, both with urea and ammonium nitrate fertilization. At 15 and 20 °C, significantly more N-NO$_3^-$ was present in the urea-fertilized soil, while at 5 and 10 °C, significantly higher N-NO$_3^-$ contents were present in the ammonium nitrate-fertilized soil. In both cases, the relationships between N-NO$_3^-$ in the 15–30 cm soil layer and the ambient temperature were strong ($R^2 = 0.753$ and $R^2 = 0.894$, respectively), but not significant ($p \geq 0.05$).

Experiment 2. N transformations in agricultural soils and leaching/migration may differ under different water regimes [8,13]. When simulating the 20 mm precipitation level in the upper (0–15 cm) soil layer in the soil temperature range of 5–20 °C, significantly more N-NH$_4^+$ was present in the urea-fertilized soil compared with the ammonium nitrate-fertilized soil (Figure 3). Similar trends were also determined at the 10 mm precipitation level (Figure 1).

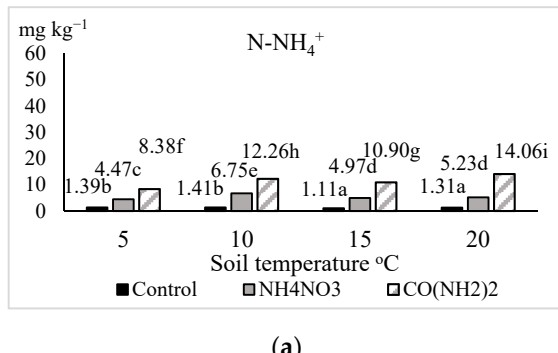 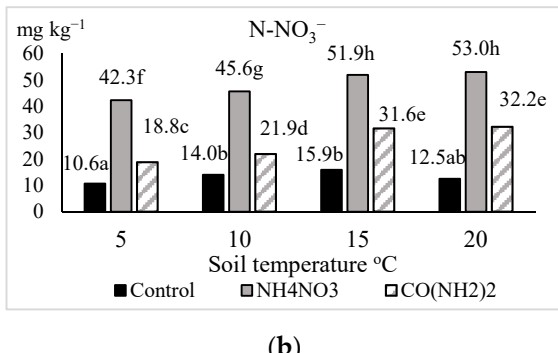

(**a**)　　　　　　　　　　　　　　　　　　　　　(**b**)

**Figure 3.** The influence of nitrogen fertilizers and ambient temperature on changes of (**a**) N-NH$_4^+$ and (**b**) N-NO$_3^-$ in the 0–15 cm soil layer with a simulated precipitation amount of 20 mm. Values followed by different letters are significantly different ($p \leq 0.05$) based on Fisher's LSD test.

The highest and most significantly different N-NH$_4^+$ content in the soil, compared to other cases, was observed in the urea-fertilized soil at 20 °C. Meanwhile, having simulated the 10 mm precipitation amount, the content of N-NH$_4^+$ in soil at this temperature was significantly lower when compared with the soil at 15 °C. In the first experiment, the reduction can be attributed to a higher possibility of nitrogen loss due to volatilization, and in the second experiment it can be attributed to the lower nitrogen loss due to volatilization, as higher precipitation amounts accelerate urea dissolution and soaking in soil [37,44]. At a soil temperature of 15 °C, the N-NH$_4^+$ content was significantly lower than at 10 °C; however, the content of N-NO$_3^-$ in the soil for this treatment was 1.2 time higher, indicating different intensities for the mineralization process. The correlation analysis of the data showed that in both urea-fertilized and ammonium nitrate-fertilized soil at the 20 mm precipitation level, the relationships of N-NH$_4^+$ with the ambient temperature were moderate ($R^2 = 0.609$ and $R^2 = 0.353$, respectively), but significant ($p \leq 0.05$), while at the 10 mm precipitation level, these relationships were very strong ($R^2 = 0.980$ and $R^2 = 0.902$, respectively) and significant ($p \leq 0.05$).

At the simulated 20 mm precipitation level, the amount of N-NO$_3^-$ tended to increase with an increasing ambient temperature, both for the urea-fertilized and ammonium nitrate-fertilized soil. The N-NO$_3^-$ content was 1.8 times lower in the urea-fertilized soil than in the ammonium nitrate-fertilized soil. In both cases, the highest N-NO$_3^-$ contents were recorded at soil temperatures of 15 °C and 20 °C and differed by a factor of 1.6 due to the form of the fertilizer. At the soil temperatures of 5 and 10 °C, the amounts of this compound differed by 2.3 times and 2.1 times, respectively.

At the 20 mm simulated precipitation level, the relationships of the N-NO$_3^-$ content with soil temperature (in the range of 5–20 °C) in the 0–15 cm soil layer for both urea-fertilized and ammonium nitrate-fertilized soil were very strong ($R^2 = 0.91$ and $R^2 = 0.96$) and significant ($p \leq 0.05$), and can be described by the following linear equations: $y_{CO(NH2)2} = 13.605 + 0.9988x$ and $y_{NH4NO3} = 38.63 + 0.767x$.

The N-NH$_4^+$ contents in the 15–30 cm soil layer varied very unevenly between both unfertilized soil and ammonium nitrate-fertilized or urea-fertilized soil. When fertilized with urea, the contents of N-NH$_4^+$ at 5 and 10 °C did not differ significantly (Figure 4). The highest content was found at a soil temperature of 15 °C, and when the temperature rose to 20 °C, the content of N-NH$_4^+$ decreased significantly. Similar changes in the N-NH$_4^+$ content in the 15–30 cm soil layer were also observed at the 10 mm precipitation level, although depending on the temperature (Figures 1 and 2).

When fertilizing with ammonium nitrate, the content of N-NH$_4^+$ was the highest at a soil temperature of 5 °C, while at soil temperatures of 10 and 15 °C, it was significantly lower. A negligible increase in this form of nitrogen was observed at 20 °C when compared with 15 °C. Such a distribution of data is explained by the regularities of soil temperature-related processes occurring in the soil [17,19]. At 5 °C, the higher N-NH$_4^+$ content in the ammonium nitrate-fertilized soil may have been caused by the slow activity of *Nitrosomonas* and *Nitrobacter* bacteria [15,16]. With an increasing soil temperature, the amount of this compound in most cases decreased, as most of it was transformed into N-NO$_3^-$;

however, a very negligible increase in the $N-NH_4^+$ nitrogen content observed at 20 °C may have been caused by the intensification of organic matter mineralization. In the urea-fertilized soil, the relationship between the $N-NH_4^+$ content in the 15–30 cm soil layer and the soil temperature was strong ($R^2 = 0.552$), but not significant ($p \leq 0.05$), whereas in the ammonium nitrate-fertilized soil, this relationship was both very strong ($R^2 = 0.902$) and significant ($p \leq 0.05$).

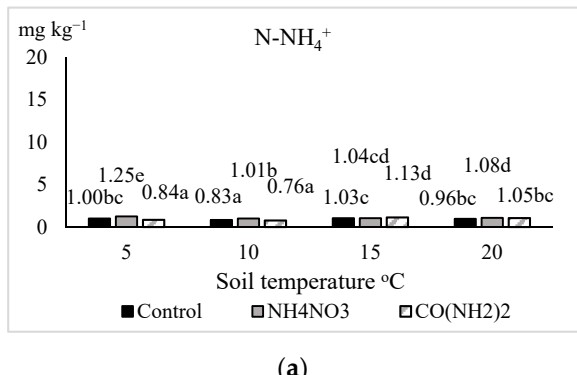 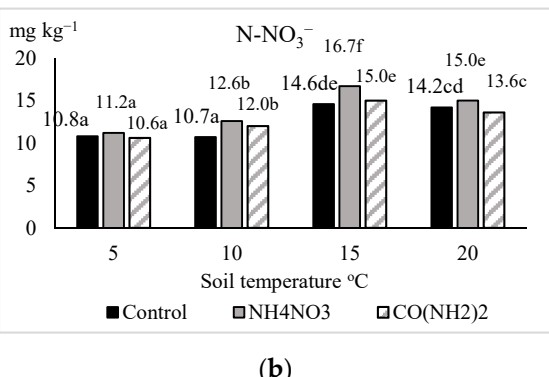

**(a)** **(b)**

**Figure 4.** The influence of nitrogen fertilizers and ambient temperature on the changes of (**a**) $N-NH_4^+$ and (**b**) $N-NO_3^-$ nitrogen in the 15–30 cm soil layer with a simulated precipitation of 20 mm. Values followed by different letters are significantly different ($p \leq 0.05$) based on Fisher's LSD test.

In both the urea-fertilized and ammonium nitrate-fertilized soil samples, significantly higher levels of $N-NO_3^-$ nitrogen in the 15–30 cm soil layer were recorded at soil temperatures of 15 and 20 °C (Figure 4). This indicates an intensive process of nitrification under these conditions, which is corroborated by the research findings of other researchers [15,16].

At lower soil temperatures (5 and 10 °C), the content of $N-NO_3^-$ in the soil was on average 21% lower in the urea-fertilized soil and 25% lower in the ammonium nitrate-fertilized soil when compared with that at 15 and 20 °C. At a simulated precipitation level of 20 mm, the content of $N-NO_3^-$ in the soil, regardless of the fertilizer applied, was about 10% higher than that at the 10 mm precipitation level.

The regression analysis showed that at the 20 mm precipitation level, the relationships between $N-NO_3^-$ in the 15–30 cm soil layer and the soil temperature (in the range of 5–20 °C) for the urea-fertilized and ammonium nitrate-fertilized treatments were both strong ($R^2 = 0.840$ and $R^2 = 0.805$, respectively) and significant ($p \leq 0.05$), and may be described by the following quadratic equations: $y_{NH4NO3} = 6.025 + 1.1035x - 0.0318x^2$ and $y_{CO(NH2)2} = 6.2625 + 0.9367x - 0.0277x^2$.

Zhang et al. [45] and Shaviv and Hagin [46] suggest that the ratio of $N-NH_4^+$ to $N-NO_3^-$ is of a great significance and can impact plant growth. The highest grain yield and total yield were obtained with a 25:100 mixture of $N-NH_4^+/N-NO_3^-$ nitrogen. The findings of the current study suggest that this ratio is closer to optimal (28–40/60–72%) with urea fertilization than with ammonium nitrate fertilization (9–17/87–91%).

The content of $N_{min}$ in the 0–15 cm soil layer was directly dependent on the fertilizer form, i.e., on their chemical composition and soil temperature at both simulated precipitation levels (10 and 20 mm). Significantly higher amounts of $N_{min}$ at the simulated 10 mm precipitation level were found in the ammonium nitrate-fertilized soil with a steady increase in soil temperature (Table 1). Depending on the soil temperature, the content of $N_{min}$ in the ammonium nitrate-fertilized soil was 1.4–1.8 times higher than that in the urea-fertilized soil. The smallest differences in the amounts of this compound resulting from different forms of fertilizers were determined at 15 and 20 °C. The lowest amounts of $N_{min}$ in the soil were determined at 5 and 10 °C.

**Table 1.** The effect of nitrogen fertilizer forms, ambient temperature, and amounts of precipitation on the changes of $N_{min}$ in the 0–30 cm soil layer.

| Fertilizer (Factor A) | Temperature (°C) in the 0–15 cm Soil Layer (Factor B) | | | | Temperature (°C) in the 0–15 cm Soil Layer (Factor B) | | | |
|---|---|---|---|---|---|---|---|---|
| | 5 | 10 | 15 | 20 | 5 | 10 | 15 | 20 |
| *Simulated precipitation at 10 mm* | | | | | | | | |
| Control | 12.0a | 13.6b | 17.3c | 20.9d | 10.7a | 11.5b | 14.6de | 14.1d |
| $NH_4NO_3$ | 46.0i | 48.7lj | 54.5k | 54.4k | 11.5b | 12.6c | 14.5de | 13.7d |
| $CO(NH_2)_2$ | 25.2e | 31.2f | 39.9h | 37.4g | 10.5a | 11.2ab | 15.3e | 14.2de |
| *Simulated precipitation at 20 mm* | | | | | | | | |
| Control | 12.0a | 15.4bc | 17.0c | 13.8ab | 11.8ab | 11.5a | 15.7ef | 15.2de |
| $NH_4NO_3$ | 46.8g | 52.4h | 56.9i | 58.2i | 12.4b | 13.6c | 17.7g | 16.0ef |
| $CO(NH_2)_2$ | 27.1d | 34.1e | 42.4f | 46.2g | 11.4a | 12.7b | 16.1f | 14.7d |

Values followed by different letters are significantly different ($p \leq 0.05$) based on Fisher's LSD test.

Several researchers, including Kandeler et al. [47], Tripolskaja et al. [48], and Zhang et al. [12], have suggested that this is related to the activity of bacteria involved in the processes of nitrogen transformation and organic matter mineralization at higher soil temperatures. This is in line with Cookson et al. [49], who presented data from laboratory research suggesting that the concentration of $N_{min}$ in soil significantly ($p \leq 0.05$) increases at a 15 °C soil temperature. In the urea-fertilized soil, the relationship between the $N_{min}$ content and ambient temperature (in the range of 5–20 °C) was described by a quadratic equation here, while in the ammonium nitrogen-fertilized soil, this relationship was defined by a linear equation. In both cases, the relationships are strong ($R^2 = 0.906$ and $R^2 = 0.711$). Urea-fertilization at temperatures above 15 °C resulted in lower $N_{min}$ contents in the soil due to nitrogen loss in the form of $N-NH_3$. This agrees with the findings of Li et al. [50], Pan et al. [51], and Rochette et al. [37], who suggest that the volatilization of $N-NH_3$ from urea has been known to be directly dependent on the ambient temperature, soil moisture, and other factors. Gardinier et al. [52] and Sommer and Hutchings [53] have also indicated that nitrogen losses in the form of ammonium begin at ambient temperatures above 15.5 °C.

The data from our study show that the content of $N_{min}$ at 5–10 °C is significantly higher in ammonium nitrogen-fertilized soil, and no significant differences caused by the fertilizer form were found at soil temperatures of 15 and 20 °C. In all cases, the amount of $N_{min}$ in the 15–30 cm soil layer was significantly lower than in the 0–15 cm layer, possibly due not only to the slow migration of $N_{min}$ resulting from the low precipitation amount, but also due to lower soil biological activity with an increasing soil depth [47].

The findings of the second experiment (simulated precipitation level of 20 mm) show that the content of $N_{min}$ in the 0–15 and 15–30 cm soil layers increased with an increasing ambient temperature for both the ammonium nitrate-fertilized and urea-fertilized soils. In both cases, the relationships between $N_{min}$ in the soil and soil temperature were strong ($R^2 = 0.782$ and $R^2 = 0.818$, respectively). Given the fact that the amount of nitrogen (as an active ingredient) applied with fertilizers was the same, the increase in $N_{min}$ in the ammonium nitrate-fertilized soil was only possible due to the mineralization of soil organic matter, whereas in the urea-fertilized soil, this increase was possible due to amide nitrogen ($N-NH_2^-$) transformation to $N-NH_4^+$. The former has been confirmed by the $N_{min}$ changes in the control treatment soil, i.e., at 20 °C, the $N_{min}$ nitrogen content was 54.4% higher than at 5 °C (estimated 0–30 cm soil layer data). The effect of soil temperature on the mineralization process in the soil has been documented by Hagemann et al. [13], Wang et al. [11], and Zhang et al. [12]. It is also important to note that when fertilizing with urea, the content of $N_{min}$ in the 0–30 cm soil layer (i.e., all considered layers) was on average 1.4 times lower when compared with the ammonium nitrate fertilization. Firstly, this could be explained by the incomplete hydrolysis of urea in 7 days and, secondly, by different mineralization intensities via the influence of different fertilizers. More comprehensive research is needed to confirm these assumptions, especially the latter one.

## 4. Conclusions

The longer that soil nitrogen can remain in the ammonium (N-NH$_4^+$) ion form, the lower the chances of nitrogen losses through leaching or denitrification are. The regularities and intensities of the transformation of amide and low-mobility ammonium nitrogen (N-NH$_4^+$) into highly mobile nitrate nitrogen (N-NO$_3^-$) affect the nitrogen fertilizer use efficiency, along with the quality of the environment, especially in the cases where the likelihood of nitrate nitrogen loss due to leaching is high and plant uptake is low.

The content of ammonium nitrogen (N-NH$_4^+$) over seven days in a soil temperature range of 5–20 °C was significantly higher in the urea-fertilized soil than in the ammonium nitrate-fertilized soil here. The highest amounts of ammonium nitrogen (N-NH$_4^+$) were recorded in the soil at 15 °C, while at 20 °C the contents significantly decreased due to more intensive nitrification processes, which resulted in significant increases in the nitrate nitrogen contents and possibly more intensive volatilization in the form of NH$_3$. When fertilizing with urea, the ratio of N-NH$_4^+$ and N-NO$_3^-$ was closer to the optimal ratio (25/75%) when compared with the ammonium nitrate fertilization. The migration of ammonium nitrogen (N-NH$_4^+$) into the deeper 15–30 cm layer under the influence of the precipitation levels of 10 and 20 mm was negligible.

The amounts of nitrate nitrogen (N-NO$_3^-$) in the 0–15 cm soil layer in the soil temperature range of 5–20 °C were 1.7–2.3 times lower when fertilizing with urea than when fertilizing with ammonium nitrate at a simulated precipitation level of 10 mm, and 1.6–2.2 times lower at a simulated precipitation level of 20 mm. With increasing soil temperatures, the amounts of nitrate nitrogen (N-NO$_3^-$) in the 0–15 and 15–30 cm soil layers increased.

**Author Contributions:** Conceptualization, R.D. and I.P.; methodology, D.J.; software, I.P.; validation, R.D., I.P, D.J., and A.P; formal analysis, I.P.; investigation, R.D.; resources, A.P.; data curation, D.J.; writing—original draft preparation, R.D. and I.P.; writing—review and editing, R.D., I.P, D.J., and A.P.; visualization, R.D.; supervision, D.J.; All authors have read and agreed to the published version of the manuscript.

**Funding:** This research received no external funding.

**Conflicts of Interest:** The authors declare no conflict of interest.

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
