# Peer review of "The Influence of Various Forms of Nitrogen Fertilization and Meteorological Factors on Nitrogen Compounds in Soil under Laboratory Conditions"

_agronomy, doi:10.3390/agronomy10122011_

Round 1

Reviewer 1 Report

  • The MS should be carefully corrected as there are quite a few grammatical errors in it. Right in the first sentence of the abstract: The most pressing environmental problems is nitrogen pollution… It should be … problem is … I try to indicate these types of errors and typos in the text with a ‘GE’ comment or with concrete suggestion.
  • How water (precipitation) was applied in the experiment?
  • 1.: It is not clear what the a, b, c… letters stand for. Please, clarify!
  • It does not influence the publication of the MS, but the Authors may have to redesign the experiment as no major difference could be detected between the two precipitation regimes. I think a 1×15 mm + 1×30 mm regime would be a better choice.
  • See other comments in the attached file.

Author Response

Dear Reviewer,

Please find enclosed our revised manuscript entitled “Effects of fertilizer forms, soil temperature and precipitation on soil nitrogen forms under controlled climate conditions” with Manuscript ID agronomy-1027881, which was modified from the original manuscript based on your instruction.

We changed the title of the manuscript according to “Reviewer 2” suggestion. “The influence of various forms of nitrogen fertilization and meteorological factors on nitrogen compounds in soil under laboratory conditions”).

Followings are revised points and some explanations:

  1. The manuscript, amended according to your suggestions, has been submitted to MDPI language editing (English editing ID: english-25014).
  2. The first sentence of the abstract (old Line 13-16) replaced by a new sentence (new Line 13-14).
  3. The distribution of simulated precipitation (water) was performed manually using a Stihl Sg 11 sprayer, having estimated the volume of water sprayed with one click. Watering was provided early in the morning. All pots received the same volume of water per event. The volume of water applied was determined according to the average meteorological data of the second to third ten-day period of March and the first ten-day period of April in the 2013-2017 period (Kaunas meteorological station, Lithuania 2013-2017), i.e., the beginning of the winter wheat and winter oilseed rape growing season in the northern part of the temperate climate zone (coordinates 54°53’3.26”, 23°50’33.25”), which is when the highest rates of nitrogen are used.
  4. The a, b, c… letters (old Line 142-143) changed to (Values followed by different letters are significantly different (P ≤ 0.05) based on Fisher’s least significant difference (LSD) test.).
  5. The experimental methodology was based on the average meteorological data, and before the experiment, it was difficult to predict whether we would get significant differences or not. Thank you for the remark and suggestions, they will be useful to us in our future research.

For the marked part in your checked document:

  1. Note, L13-16: The first sentence of the abstract replaced by a new sentence (Nitrogen is one of the main factors that shapes soil fertility and the productivity of crops, although its abundance can cause also damage to the environment [1].).
  2. Note, L16: The word “the“ of the abstract deleted.
  3. Note, L19: The word “transformation“ of the abstract deleted.
  4. Note, L32: The first sentence of the introduction deleted.
  5. Note, L39: The word “reducing“ changed to “reduce“.
  6. Note, L48-49: The sentence of the introduction deleted.
  7. Note, L55-57: The sentence replaced by a new sentence (Tripolskaja and VerbylienÄ— [24] suggest that precipitation during the warm period of the year promotes the leaching and migration of chemical elements from topsoil.).
  8. Note, L58: The word “most“ deleted.
  9. Note, L60: The word “favourable“ changed to “certain“.
  10. Note, L 64, 65: The word “hereinafter” deleted.
  11. Note, L68: The words (N-NH4+; N-NO3- and Nmin) deleted.
  12. Note, L89: The contents of N-NH4+ and N-NO3-
  13. Note, L124: The sentence replaced by a new sentence (In the 0–15 cm soil layer, under the simulated precipitation level of 10 mm, the experimental findings show significantly higher amounts of N-NH4+ nitrogen in the urea-fertilized soil compared with the ammonium nitrate-fertilized soil (Fig. 1).).
  14. Note, L130: The word “from” changed to “about”.
  15. Note, L131: The word “temperatur” changed to “temperature” and the word “ambient” deleted.
  16. Note, L142: The a, b, c… replaced by a new sentence (Values followed by different letters are significantly different at (P≤0.05) based on Fisher’s LSD test).
  17. Note, L145: The word “change” changed to “difference”.
  18. Note, L157: The number “1“ and dash “-“ deleted.
  19. Note, L161: The title of analysis “correlation regression” changed to “regression”.
  20. Note, L162: We agree with your opinion.
  21. Note, L164: The comma changed to a full stop.
  22. Note, L168: The word “the” deleted.
  23. Note, L185: This sentence deleted and “sorb” changed to “absorb”
  24. Note, L215: The word “amount” deleted.
  25. Note, L216: The word “significant” changed to “significantly different”.
  26. Note, L217: The word “established” changed to “observed”.
  27. Note, L222: The word “uptake” changed to “soaking”.
  28. Note, L227: The word “and” changed to “but”.
  29. Note, L305: We added an extra line here.

We are extremely grateful to the Reviewer for highlighting all these shortcomings (substantive and language ones as well as technical errors) in our paper. We appreciate the Reviewer’s enormous effort and time devoted to such an in-depth and thorough analysis of this paper. We are convinced that all comments of the Reviewer have produced a much better technical quality of our paper. We hope that after making revisions in it, the paper meets the requirements for publication in Agronomy.

We kindly ask you to publish this paper in Agronomy.

Kind regards,

Assoc. prof. dr. Darija JodaugienÄ—

Reviewer 2 Report

The study assessed the effect of various forms of nitrogen fertilisation (urea, ammonium nitrate) on changes in nitrogen compounds (N-NH4, N-Min, N-NO3) in two soil layers. Nitrogenous forms were subjected to simulated rainfall (10 mm and 20 mm) and air temperature (5-20 0C) in laboratory conditions.

The article is supplemented with four figures and one table.

For the experiments carried out in the laboratory – the pot conditions, the materials and the methodology used are appropriate.

General comments to the article:

  • Failure to specify the type of soil, whether mineral or organic soil, which was used in the experiments and according to which parameters it was determined,
  • Why was the artificial rainfall limited to doses of 10 and 20 mm? What was it related to?
  • What was the intensity of the hydration? With what tools?
  • What was the soil moisture level before and after the irrigation?
  • What was the infiltration capacity of the soil used in the experiments?
  • Please formulate and list more detailed conclusions.
  • The authors of the study should relate the obtained results and conclusions to agricultural practice, or whether the conclusions from the work are applicable only in laboratory and greenhouse tests conditions.

I propose to change the title of the work to "The influence of various forms of nitrogen fertilisation and meteorological factors on nitrogen compounds in soil under laboratory conditions".

Author Response

Dear Reviewer,

Please find enclosed our revised manuscript entitled “Effects of fertilizer forms, soil temperature and precipitation on soil nitrogen forms under controlled climate conditions” with Manuscript ID agronomy-1027881, which was modified from the original manuscript based on your suggestions.

Below are revised points and some explanations:

  1. As asked, we included the description of soil type (The soil from the 0–30 cm layer of Luvisol [WRB, 214]), characterized by a loam texture (clay 15.1 %, silt 44.5 % and sand 40.4 %) was used.).
  2. We took your comments into account and explained why we used artificial rainfall limited to doses of 10 and 20 mm and the intensity of the hydration. (The distribution of simulated precipitation (water) was performed manually using a Stihl Sg 11 sprayer, having estimated the volume of water sprayed with one click. Watering was provided early in the morning. All pots received the same volume of water per event. The volume of water applied was determined according to the average meteorological data of the second to third ten-day period of March and the first ten-day period of April in the 2013-2017 period (Kaunas meteorological station, Lithuania 2013-2017), i.e., the beginning of the winter wheat and winter oilseed rape growing season in the northern part of the temperate climate zone (coordinates 54°53’3.26”, 23°50’33.25”), which is when the highest rates of nitrogen are used.).
  3. We took into account your comments regarding soil moisture level before and after the irrigation. (The soil moisture levels were the following: 1) before the first irrigation—25.0%; after the first irrigation—26.7% (precipitation of 10 mm) and 28.3% (precipitation of 20 mm); 2) before the second irrigation—24.0–26.0% (precipitation of 10 mm) and 25.6–27.6% (precipitation of 20 mm); after the second irrigation—25.7–27.7% (precipitation of 10 mm) and 28.9 –30.9% (precipitation of 20 mm).).
  4. The infiltration capacity of the soil in the experiments was not determined and therefore cannot be provided.
  5. New conclusions:

The longer that soil nitrogen can remain in the ammonium (N-NH4+) ion form, the lower the chances of nitrogen losses through leaching or denitrification are. The regularities and intensities of the transformation of amide and low-mobility ammonium nitrogen (N-NH4+) into highly mobile nitrate nitrogen (N-NO3-) affect the nitrogen fertilizer use efficiency, along with the quality of the environment, especially in the cases where the likelihood of nitrate nitrogen loss due to leaching is high and plant uptake is low.

The content of ammonium nitrogen (N-NH4+) over seven days in a soil temperature range of 5–20 °C was significantly higher in the urea-fertilized soil than in the ammonium nitrate-fertilized soil here. The highest amounts of ammonium nitrogen (N-NH4+) were recorded in the soil at 15 °C, while at 20 °C the contents significantly decreased due to more intensive nitrification processes, which resulted in significant increases in the nitrate nitrogen contents and possibly more intensive volatilization in the form of NH3. When fertilizing with urea, the ratio of N-NH4+ and N-NO3- was closer to the optimal ratio (25/75%) when compared with the ammonium nitrate fertilization. The migration of ammonium nitrogen (N-NH4+) into the deeper 15–30 cm layer under the influence of the precipitation levels of 10 and 20 mm was negligible.

The amounts of nitrate nitrogen (N-NO3-) in the 0–15 cm soil layer in the soil temperature range of 5–20 °C were 1.7–2.3 times lower when fertilizing with urea than when fertilizing with ammonium nitrate at a simulated precipitation level of 10 mm and 1.6–2.2 times lower at a simulated precipitation level of 20 mm. With increasing soil temperatures, the amounts of nitrate nitrogen (N-NO3-) in the 0–15 and 15–30 cm soil layers increased.

  1. We changed the title of the manuscript according to your suggestion. ("The influence of various forms of nitrogen fertilization and meteorological factors on nitrogen compounds in soil under laboratory conditions").

We are extremely grateful to the Reviewer for highlighting all the shortcomings in our paper. We appreciate the Reviewer’s enormous effort and time devoted to such an in-depth and thorough analysis of this paper. We are convinced that all the comments of the Reviewer have produced a much better technical quality of our paper. We hope that after making revisions in it, the paper meets the requirements for publication in Agronomy.

We kindly ask you to publish this paper in Agronomy.

  Kind regards,

Assoc. prof. dr. Darija JodaugienÄ—

Round 2

Reviewer 2 Report

The article may be published in its current form, after taking into account the introduced amendments.